# Changing parental feeding practices through web-based interventions: A systematic review and meta-analysis

**Ana Isabel Gomes**[1‡]*, **Ana Isabel Pereira**[1‡], **Magda Sofia Roberto**[1‡], **Klara Boraska**[1,2‡], **Luisa Barros**[1‡]

**1** Faculty of Psychology, Research Center for Psychological Science (CICPSI), University of Lisbon, Lisbon, Portugal, **2** Department of Psychology, Catholic University of Croatia, Zagreb, Republic of Croatia

‡ AIG, AIP, MSR and LB are Senior authors. KB is a Junior author.
* ana.fernandes.gomes@psicologia.ulisboa.pt

**Data Availability Statement:** All relevant data are within the manuscript and its Supporting Information files.

## Abstract

Web-based parent interventions designed to promote children's healthy eating patterns can enhance parents' engagement and facilitate behavior change. However, it is still unclear how much the existing programs focus on changing parental feeding practices, and if so, which behavioral methodologies are used and how effective these interventions are in changing these parental behaviors. This systematic review and meta-analysis studied randomized controlled trials of web-based interventions targeting parents of 0-12-year-old children, aiming to promote children's healthy diet or prevent nutrition-related problems and reporting parental feeding behaviors as one of the outcomes. We conducted an electronic search in four databases from the earliest publication date until February 2020. Of the 1271 records found, we retained twelve studies about nine programs, comprising 1766 parents that completed the baseline evaluation. We found recent interventions, mainly directed to parents of young children, with small, non-clinical samples, and mostly theory-based. The programs were heterogeneous regarding the type of intervention delivered and its duration. The most assessed parental feeding practices were *Restriction*, *Pressure to eat*, and *Food availability/accessibility*. The behavior change techniques *Instruction on how to perform the behavior*, *Demonstration of the behavior*, and *Identification of self as role model* were frequently used. Meta-analytic results indicated that most programs' effects were small for the evaluated parental practices, except for *Food availability/accessibility* that benefited the intervention group only when all follow-up measurements were considered. The development of high-quality and controlled trials with larger samples is needed to determine with greater certainty the interventions' impact on parental feeding behaviors. The more frequent inclusion of measures to evaluate parental practices to support children's autonomy and of self-regulatory strategies as intervention components should be considered when designing programs.

**Funding:** This work was supported by the Fundação para a Ciência e Tecnologia (FCT) (https://www.fct.pt/), PTDC/PSI-GER/30432/2017. The funder had no role in study design, data collection and analysis, decision to publish, or preparation of the manuscript.

**Competing interests:** The authors have declared that no competing interests exist.

## Introduction

A healthy diet in the first years of life contributes to the development of healthier food preferences and habits that can endure through adulthood [1], thus reducing the risk for several health and development problems (e.g., obesity) [2]. Caregivers can shape children's preferences and food acceptance through positive modeling, early exposure to different healthy foods and appropriate guidance of children's eating behaviors [3, 4]. The ability to impact a child's dietary patterns is maximized in early childhood and can be extended as long as caregivers have the primary control over children's food environment [5, 6].

Parental feeding practices have been described as the specific behaviors, actions, and strategies implemented to, intentionally or not, modify children's eating behaviors [7, 8]. Parenting feeding practices can be organized in three higher-order constructs [8, 9]: i) *coercive control* (specific parent-centered strategies intended to change children's eating behavior through dominance, intrusiveness, reinforcement or supervision, e.g., pressure to eat); ii) *structure* (when parents define a set of rules and boundaries and organize the home environment to facilitate or reduce specific children's eating behaviors, e.g., food availability); and iii) *autonomy support or promotion* practices (parents' actions that support the child's initiatives and autonomy, or help the child to develop age-appropriate eating self-regulation skills, e.g., nutrition education).

The distinction between overt restrictive feeding practices (that do not consider the child's needs in the decisions about food and eating) and covert, non-coercive restrictions (that reflect parent's actions to change the food environment and set rules with the child's involvement) [8, 9] is relevant, considering the different implications of the two practices on children's dietary intake and self-regulation development [10]. Coercive controlling practices can lead to unintended consequences on children's dietary intake, contrary to parents' purposes [11]. For instance, pressuring to eat was related to higher unhealthy and lower healthy food consumption in younger children [12]. On the other hand, structuring practices like modeling and food availability were positively correlated with both children's healthy and unhealthy food intake: children tend to eat more of what their parents eat and what is available at home, independently of the type of food [12]. Positive parental feeding practices that are simultaneously high on demandingness and responsiveness can contribute to children's healthier dietary and weight outcomes across childhood and adolescence [13].

Interventions that use the internet and mobile devices or, more recently, app functionalities [14] to change children's diet through parent feeding behaviors, have raised specialists' interest. These interventions have an appealing and interactive way of delivering information, offer individually tailored feedback and strategies, resulting in more flexible programs that are cost-effective and have a higher reach [15, 16]. Considering that most face-to-face parent interventions have low retention and high dropouts [17], this approach has great potential. A recent study found that 54.8% of parents of young children prefer to develop strategies to deal with their child's feeding difficulties through online programs, exclusively or combined with face-to-face interventions. Moreover, parents indicate that they are willing to participate in longer online interventions (12 weeks) comparing with face-to-face interventions (4 weeks) [18]. Thus, web-based programs appear to be a promising strategy to change health-related parenting behaviors [19, 20].

Earlier reviews on general interventional studies have pointed to the effectiveness of parental feeding interventions on children's dietary behaviors. For children up to five years, interventions that target child feeding practices (e.g., repeated exposure, flavor-flavor learning, prompting, modeling) effectively increase fruit and vegetable intake with effects lasting up to 12 months [21]. Interventions with preschool children that included repeated taste exposure

and reward had better results regarding vegetable intake than those that did not [22]. For children under three, exposure (with or without consumption) to various and unfamiliar vegetables also diminished the reluctance to try a new vegetable [23]. However, less is known about the specific parental feeding practices that changed as a result of the intervention. The available reviews of programs to promote children's healthy eating have a broad scope regarding the type of intervention delivered and the outcomes considered for analysis [21, 24]. Reviews focused on technology-based programs do not explicitly define parents as the main targets of the intervention [25, 26] and include interventions combining online and face-to-face modules [27, 28]. In either case, the impact of the programs on parental feeding practices is not assessed.

The enhancement of behavior change programs may depend on finding the best approaches to promote positive parental feeding practices and help parents overcome obstacles that may discourage or hinder these behaviors [29, 30]. There is some information about the components used in parenting interventions to improve children's diet. Behavior change techniques like instructions on how to perform the behavior, social support, role modeling, and goal setting are frequently used in parental interventions to reduce children's intake of unhealthy foods [31]. Effective interventions on promoting weight-related nutrition intake in children and adolescents share similar components like prompting participants to self-monitor, restructure the home food environment, or identify barriers to a healthy diet [32]. However, a specific analysis of behavior change techniques used to change parental feeding practices in web-based interventions that target these practices as outcomes is still missing. This analysis can help draw conclusions about which mediating processes are most effective in changing parental feeding practices and allow program designers to make better-informed decisions [33].

The current systematic review and meta-analysis of parental web-based interventions to change parental feeding practices in the context of healthy eating promotion addresses the following questions: a) Which parental feeding practices are studied as outcomes of the programs?; b) Which behavior change techniques are used to promote changes in parental feeding practices?; c) What is the effectiveness of these interventions on changing the different parental feeding practices?

## Materials and methods

The current systematic review and meta-analysis followed the Preferred Reporting Items for Systematic Reviews and Meta-analysis (PRISMA) guidelines [34]. The completed PRISMA checklist is available in S1 File.

### Search strategy and selection criteria

Several inclusion criteria were considered for this review. The studies were eligible for analysis if: a) it targeted parents of children between 0 and 12 years old; b) the intervention aimed to promote children's healthy diet and/or to prevent nutrition-related problems (e.g., excessive weight) both in healthy or clinical populations; c) it included a web-based intervention (e.g., interactive and computerized resources delivered through the internet, websites, serious games, and app functionalities) as a stand-alone intervention for parents; we included studies where children also received some intervention, as long as the parents were the main target of the intervention, and this intervention was exclusively web-based; d) parental feeding practices were one of the outcomes; e) parental feeding practices were assessed through quantitative measures (e.g., self-report questionnaires); f) it used a randomized controlled trial (RCT) design. We excluded the studies: a) where the intervention was directed only to parents of

adolescents; b) that did not include published results on parental feeding practices outcomes (e.g., study protocols); d) that were reviews or meta-analysis and; e) were not in the English language.

To carry out a comprehensive search, we included four databases (SCOPUS, Web of Science, EBSCO, and CENTRAL) using a systematic search protocol and keywords defined by specific criteria according to the PICOS approach (Table 1) [34, 35]. According to the study's objectives, the search terms were defined after consulting the MeSh terms, several papers, and other reviews about technology-based nutritional interventions. The search was run for titles, keywords, and abstracts. We also searched for other intervention studies that complied with these criteria in the reference section of published review papers. Relevant study protocols and clinical trials were traced to find further publications of the results of the intervention.

The electronic database search was conducted on December 9th, 2019, and updated on February 13th, 2020. The records were exported to Zotero®, where the duplicates were removed, and the first phase of the selection was done (e.g., records screened by title and abstract) (Fig 1). A two-phase procedure was used in which first titles/abstracts and then full articles were screened. Both steps of the studies' selection were conducted independently by two authors (A. I.G. and L.B.) according to the defined inclusion/exclusion criteria. When disagreements occurred, a third author (A.I.P.) was consulted to reach a final decision by consensus.

## Data extraction

We defined a protocol to extract relevant data about the studies, regarding: a) study identification (e.g., authors, year of publication, country, and study design); b) objectives; c) target population (e.g., number of participants and characteristics, attrition/dropout rate); intervention and control conditions (e.g., samples, type of intervention/control condition, contents, duration, and regularity); f) variables of the study (e.g., specific parental feeding practices and other variables measured, assessment timings); and g) outcome data. Two authors (A.I.G. and K.B.) collected the information, and a third author (A.I.P.) revised it. When the information required was not thoroughly described in the article, we contacted the authors.

## Risk of bias assessment

The risk of bias was assessed using the RoB2, a revised Cochrane tool that provides a framework to evaluate the risk of bias in RCTs [37]. The five domains of the tool (e.g., *1. Bias arising*

**Table 1. Protocol and items applied in the systematic review search (PICOS approach).**

| PICOS | Search items |
|---|---|
| Population, patient or problem | Parent* OR mother* OR father* OR caregiver* OR caretaker* OR family |
| | AND Child* OR preschool* OR toddlers OR infants |
| | AND Obesity prevention OR healthy eating promotion OR fruit* OR vegetable* OR sugar* OR meal* |
| | NOT |
| | Adolescence OR adolescents OR teens OR secondary students OR secondary school |
| Intervention | Online OR web OR mHealth OR eHealth OR mobile OR application OR computer OR smartphone |
| Comparator | Control group OR treatment as usual |
| Outcome | Feeding practices OR feeding strategies OR feeding habit* OR feeding behav* |
| Study design | Intervention OR trial OR program* OR effectiveness OR efficacy OR randomized controlled trial OR RCT |
| | NOT |
| | Review OR meta-analysis OR systematic review |

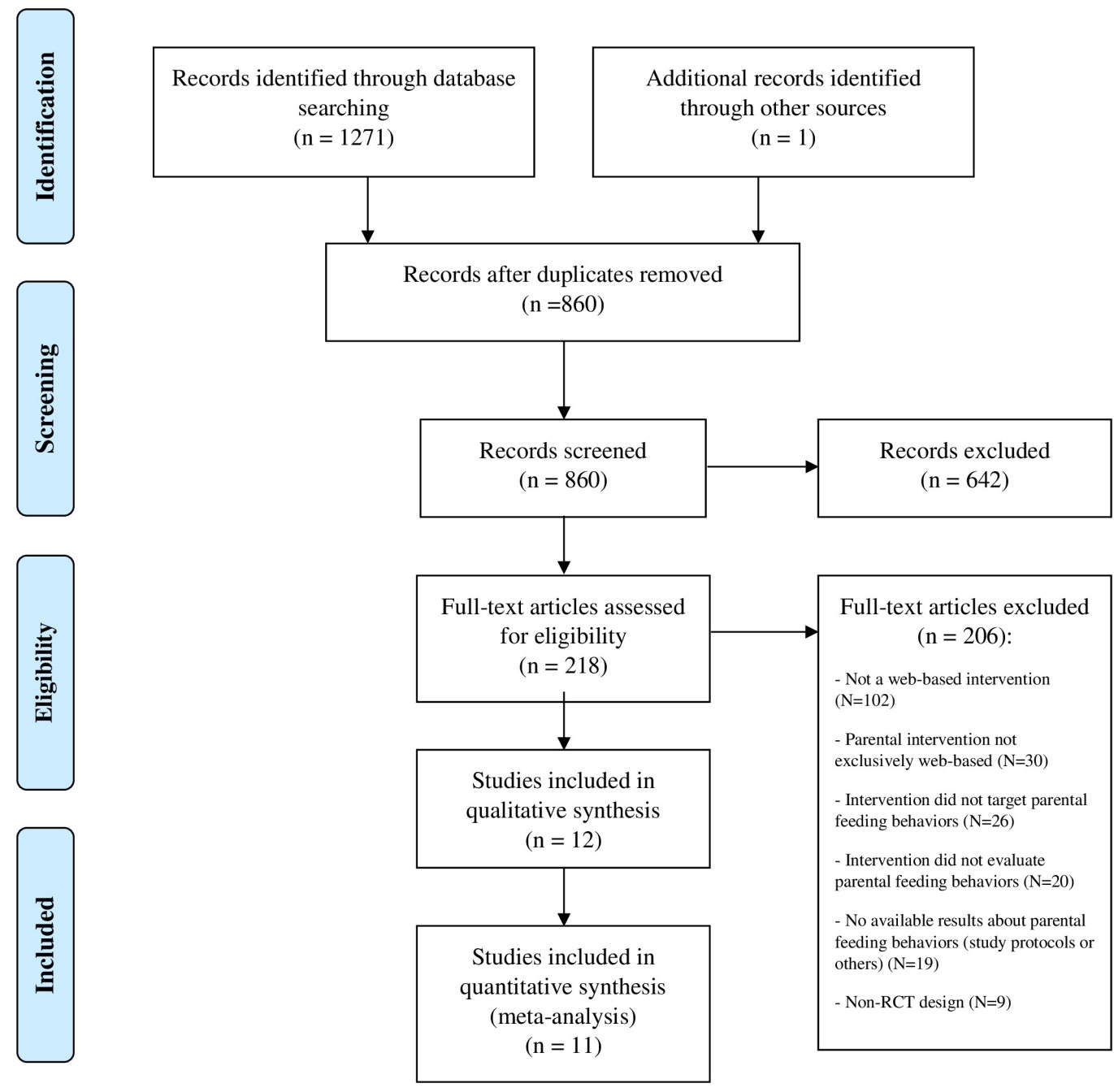

**Fig 1. PRISMA flow diagram about the study selection process [36].**

*from the randomization process; 2. Bias due to deviations from intended interventions; 3. Bias due to missing outcome data; 4. Bias in the measurement of the outcome; and 5. Bias in the selection of the reported result*) correspond to the main types of bias that can influence the study results. Because RoB2 is a results-based tool, we evaluated the bias for the results reported in each study (related to measurements of the same dimension in different time points, e.g., immediately after the intervention, three months follow-up, one-year follow-up), and not for the whole RCT [37]. For each domain, two authors (A.I.G. and L.B.) responded independently

to the signaling questions regarding risk-of-bias judgments. In the case of rating conflicts, both authors discussed and reached a consensus, helped by a third author (A.I.P.).

## Parental feeding practices and behavior change techniques codification

Considering that we aimed to identify which parental feeding practices were targeted as outcome variables and evaluate how effective these interventions were in changing those feeding practices, we focused on the instruments used to assess parenting practices as outcomes in each trial. All articles were coded for the type of parental feeding practice outcomes measured. The codification of the parental feeding practices outcomes was based on the conceptual framework proposed by O'Connor et al. [9], also considering earlier contributions of Vaughn et al. [8]. We considered three major parental feeding practices categories: *Coercive control*, *Structure*, and *Autonomy support and promotion*.

To identify and code the behavior change techniques (BCT) applied in the interventions, we used Michie's et al. [38] taxonomy (BCTTv1), according to the six coding principles and the description/examples given by the authors for each BCT (e.g., online training in BCTTv1 use, www.bct-taxonomy.com). This taxonomy includes 93 BCTs, organized in 16 categories or clusters (*1. Goals and planning, 2. Feedback and monitoring, 3. Social support, 4. Shaping knowledge, 5. Natural consequences, 6. Comparison of behavior, 7. Associations, 8. Repetition and substitution, 9. Comparison of outcomes, 10. Reward and threat, 11. Regulation, 12. Antecedents, 13. Identity, 14. Scheduled consequences, 15. Self-belief, 16. Covert learning*). Both classifications were based on the interventions' descriptions, considering the articles selected for the review and other published studies about the program (e.g., trial registries, protocol studies, dissertation thesis, feasibility studies). The BCT taxonomy was applied to the intervention actions exclusively directed to parents, regarding the nutritional component only, and with parental feeding practices as target behavior. Both categorizations were performed independently by two authors (A.I.G. and L.B.) and discussed with a third author (A.I.P.) before a final decision was made.

## Data analysis and synthesis

For the meta-analysis, one program (*Project FUN)* was excluded due to the type of analysis performed (i.e., parental feeding practices were examined as predictors of children's BMI changes after the intervention). Then, we checked for parental feeding dimensions that were only assessed by one of the programs (i.e., *Threats and bribes*, *Intrusive control*, *Prompt to eat*) and excluded these dimensions from the analysis. For each of the remaining dimensions, we verified the type of variable considered (i.e., nominal, ordinal, interval, ratio) and retained only the programs for which there was a mean score for each of the parental feeding outcomes (mandatory for performing this statistical analysis).

Finally, eleven studies, describing eight programs, were pooled for the meta-analysis, which comprised 68 effects grouped by time points (from post-intervention to additional follow-up measurements) according to the following dimensions: *Restriction*, *Pressure to eat*, *Food to control negative emotions*, *Monitoring*, *Meal and snack routines*, *Modeling*, *Food availability/ accessibility*, *Food preparation* and *Encouraging healthy eating*. An additional categorization was performed to clarify effects for *Meal and snack routines* and *Food availability/accessibility*, with effects grouped as positive or negative routines and healthy and unhealthy food availability.

Because different publications reported on the results from the same program, data were analyzed considering programs. Thus, for each program, means and standard deviations for the intervention and control groups and their sample size were retrieved to compute Hedges *g*,

the standardized effect size measure used to ensure comparability between outcomes and different sample sizes [39]. Weighted effect sizes considering within-effects dependency were estimated: first, effects from all time-points by dimension were used; afterward, only effects at post-intervention (T2) were included. The former allowed us to consider all information available, while the latter contributed to clarifying the estimates by focusing on the time point with more reported data. The overall treatment effects are estimated for each specific parental feeding practice, considering the heterogeneity of these dimensions. We used a random-effects model based on the robust variance estimation approach (RVE) to account for intra-study dependence and provide estimates corrected for small samples [40]. For each effect ($g$), a 95% confidence interval (CI) was calculated. For significant effects, intervals did not include 0. The $I^2$ statistic was used to evaluate heterogeneity between studies (low heterogeneity $I^2 < 50\%$, moderate heterogeneity $I^2$ values between 50% and 75%, high heterogeneity, $I^2 > 75\%$) [41]. To report on the expected lower and upper limits for future effects, we computed prediction intervals [42]. Forest plots represented the average weighted effects.

R software [43] and the following packages were used for the analysis: the *esc* package [44] for effects conversion, the *robust* package [45] for estimating the effects following RVE, and the *meta* [46] and *metafor* [47] packages for graphical representations.

## Results

Of the 1271 records found (Fig 1), 12 articles [48–59] met all the defined criteria and represented nine programs. The results of two programs, *EMPOWER* [56–58] and *Early Food for Future Health* [49, 50], were reported on more than one study considering different follow-up measurements. All the studies were included in the systematic review, and 11 were retained for the meta-analysis (Fig 1). The information about the articles selected for review is summarized in the S1 Table.

### Risk of bias

Detailed information about the risk of bias judgment for each study (final decision) is available in the S2 File. The overall risk of bias was *high* for seven studies, three of which refer to the *EMPOWER* program (Fig 2). The studies describing the results for *Early Food for Future Health* and *HomeStyles* programs also raised *some concerns*.

Five studies were judged to be adequate (*low risk of bias*) regarding the *Randomization process*. Although the sequence generation was reported for most studies, the information about the allocation sequence's concealment was missing. The risk of bias was considered *high* when the research team or one of the investigators performed the participants' allocation (three studies). The risk of bias for *Deviations from intended interventions* was considered *low* for all studies. Although most studies did not provide clear information about blinding, the interventions were delivered online, according to a specific protocol, without the researchers' interference. Moreover, all studies used intention-to-treat (ITT) or modified intention-to-treat (mITT) analyses, which was considered appropriate. Regarding *Missing outcome data*, only one study raised *some concerns* due to the absence of a flow chart and documented reasons for withdrawal. The major issues were identified in the *Measurement of the outcome* domain: four studies were judged to have a *high risk of bias*, and two studies about the same program raised *some concerns*. In non-blinded studies or in studies where the information about blinding was missing, it was impossible to guarantee that the knowledge about the different conditions did not influence their answers, since the participants were the outcome assessors for their feeding practices. This risk of bias was considered potentially higher when the control and intervention conditions were quite different regarding its contents, structure, or degree of participants'

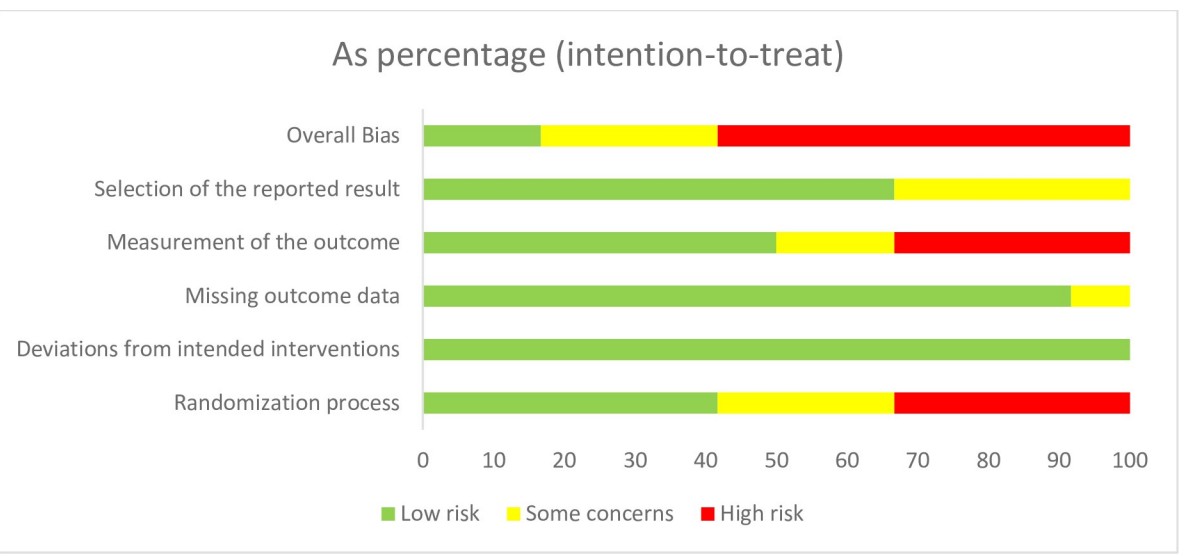

**Fig 2. Judgments about the risk of bias for each dimension (RoB2 tool) [60] presented as percentages across studies.**

involvement. This domain was not affected by the researchers' knowledge about the participants' allocation since they had no active participation in the intervention and could not change the intervention procedures (i.e., online intervention). Four studies raised *some concerns* regarding the *Selection of the reported results* because the researchers' pre-specified intentions and analysis plan (i.e., at the trial protocol) were not available.

## Study and sample characteristics

The 12 articles were published between 2013 and 2019 (50% in 2018 or 2019), mainly by research teams from the United States (eight articles). The information about the theoretical frameworks used to develop the intervention and study design was available for eight programs. Most programs were based on Social Cognitive Theory (SCT) (six programs). Trials also mentioned other models, independently or complementary to SCT: attachment and anticipatory guidance models [49], the Theory of Planned Behavior [53], Information-Motivation-Behavior Model [51], Socio-Ecological Model, Adult Learning Theory, and Motivational Interviewing [59].

Five programs also focused on other topics besides children's nutrition, e.g., physical activity [48, 54, 55, 57, 59], sleep [48, 59] and screen time [48, 54, 57]. The percentage of sessions or modules dedicated to nutritional issues ranged from 33.3% to 83.3% (M = 58.04, SD = 18.37) in those programs. Six interventions targeted parents of 2 to 6 years old children exclusively. Most interventions focused on community samples, and two considered specific inclusion criteria regarding children's BMI percentile (e.g., at or above the WHO 50th BMI percentile for age and sex) [48] or parents' BMI and waist circumference (e.g., BMI equal or higher than 23 or waist circumference higher than 31,5 inches) [51].

All programs, except one [55], used a two-arm design and most trials offered some kind of intervention in the control condition. Three programs [48, 57, 59] had an attention control or active comparator group, similar to the intervention condition, but without the 'active ingredient'. In these cases, the control condition maintained the structure and other non-specific treatment effects but changed the intervention's contents or was mainly a knowledge-based intervention about the same themes without the theory-based methodologies that were

considered essential for behavior change. The control group of another four programs [51–54] provided a minimal intervention, with a lower participant burden than the experimental condition, and included contents unrelated to the main program. The remaining programs used the treatment as usual [49] or evaluation only [55] as control conditions.

The programs lasted from 1 [52] to 36 weeks [53, 59] (M = 14.44; SD = 12.85). The studies included one [52, 55, 59], two [48, 50, 51, 53, 54], or four [58] post-intervention follow-up measurements. Study durations ranged from 1 [52] to 118 weeks [58] (M = 44.17; SD = 31.64).

Although the interventions were exclusively web-based, five included direct contacts with participants, for assessment purposes [51, 55], feedback or nudges delivery [48, 59], answer to questions and doubts [48, 51], or to encourage participation in the study [53, 55, 59]. Three programs [48, 55, 57] promoted interactions with other parents who participated in the intervention through discussion boards, live chats, or forums. Six programs included a monetary incentive at the end of the study [52, 55, 57] or after specific tasks were accomplished [48, 51, 59].

Overall, the studies comprised 3162 eligible participants, of which 1766 completed the baseline evaluation (range of sample size: 32–718 participants; M = 196.22; SD = 240.66). The average rate of participants' retention from baseline to post-intervention evaluation (considering the first follow-up measurement immediately after the intervention) was 79.2% (SD = 19.90; range: 35.2%-100%). Information about the parents' sample at baseline was available for 1382 participants [48, 49, 51, 57, 59]. Most parents were mothers (97.3%), and the mean age was 33.96 years (SD = 2.41).

## Parental feeding practices used as outcomes in the studies

Information about the instruments used to assess specific parental feeding practices (final decision) is available in the S2 Table. One program considered parental feeding practices as primary outcomes among other nutrition-related measures [49], three classified these parental dimensions as secondary outcomes [48, 57, 59], and the remaining five programs did not label the parental feeding practices as primary or secondary outcomes. Parents were the sole informants regarding their feeding practices, except in the two programs that targeted parents of older children [54, 55]. In these cases, children completed a similar version of the parents' scales or were the only evaluators of a specific parental practice. The Child Feeding Questionnaire (CFQ) was the most used instrument to assess parental feeding practices [48, 50, 51, 53]. Other questionnaires were also included, alone or combined with the CFQ (i.e., Food/Activity Parenting Practices Questionnaire, The Family Eating and Activity Habits Questionnaire, Parent Feeding Practices Scale, Infant Feeding Questionnaire, Parent Modelling Questionnaire, Family Support). Three programs developed specific instruments to evaluate parental feeding practices for the RCT, based on earlier literature research and previous measures developed by the program researchers [54, 57, 59].

Table 2 presents the categorization of parental feeding practices considered as outcome measures. All the interventions evaluated one or more parental feeding practices included in the *Structure* category, and 66.7% of the interventions measured at least two categories of parental feeding practices. The *Autonomy support and promotion* category was the least frequently evaluated in the programs. Fourteen of the 19 parental practices were measured, ranging from 1 to 8, with an average of 4.44 (SD = 2.24) practices evaluated per intervention. *Restriction*, *Pressure to eat*, and *Food availability/accessibility* were the parental practices most often measured in the interventions (> 50% interventions). Three *Structure* practices (*Rules and limits*, *Exposure to variety/selection*, *Permissive feeding*) and two *Autonomy support and promotion* practices (*Nutrition education/reasoning*, *Child involvement*) were not evaluated in any program.

**Table 2. Classification of parental feeding practices considered as outcome measures across trials.**

| | HomeStyles | Family Eats | Feeding Healthy Food to Kids | Project FUN | EMPOWER | Time2bHealthy | Early Food For Future Health | Happier Meals | 5-4-3-2-1-0 Program | Sum |
|---|---|---|---|---|---|---|---|---|---|---|
| **1. Coercive control** | | | X | X | | X | X | X | X | **6** |
| Restriction [a] | | | x | x[b,1] | | x | x | x | x | 6 |
| Pressure to eat | | | x | x[b,1] | | x | x | x | x | 6 |
| Threats and bribes | | | | | | | | x | | 1 |
| Using food to control negative emotions | | | | | | | x | x | | 2 |
| Intrusive control | | | | | | | | x[d] | | 1 |
| **2. Structure** | X | X | X | X | X | X | X | X | X | **9** |
| Rules and limits | | | | | | | | | | 0 |
| Prompt to eat | | | | x[c,2] | | | | x | | 2 |
| Monitoring [a] | | | x | x[1] | | | | x | x | 4 |
| Meal and snack routines | x | | | | | | x | | x | 3 |
| Modeling | x | x[2] | | x[c,2] | | x | | | | 4 |
| Food availability/accessibility | x | x[1] | | x[c,2] | x | | | x[d] | x | 6 |
| Food preparation | x | x | | | | | | | | 2 |
| Exposure to variety/selection | | | | | | | | | | 0 |
| Negotiation and redirection | | | | | | | | | | 0 |
| Permissive feeding | | | | | | | | | | 0 |
| **3. Autonomy support or promotion** | | | | X | | | X | | X | **3** |
| Nutrition education/reasoning | | | | | | | | | | 0 |
| Child involvement | | | | | | | | | | 0 |
| Encourage healthy eating | | | | | | | x | | x | 2 |
| Praise [a] | | | | x[c,2] | | | | | | 1 |
| ***Sum (of the feeding practices categories)*** | 1 | 1 | 2 | 3 | 1 | 2 | 3 | 2 | 3 | **18** |
| ***Sum (of the specific feeding practices)*** | 4 | 3 | 3 | 7 | 1 | 3 | 5 | 8 | 6 | **40** |

[a] *Restriction*, *Monitoring*, and *Praise* were maintained as defined in the initial content map for food parenting practices [8] to fit better all the dimensions assessed in the studies.

[b] The data related to these outcomes were collected but not included for analysis in the article due to insufficient psychometric features of the scale.

[c] The practices of *Modeling*, *Food availability/accessibility*, *Prompt to eat*, and *Praise* were assessed in this study with a 4-item scale (one item for each parental practice), and a total mean score was calculated for analysis in a dimension defined as *Family support*.

[d] Overt and covert control practices were assessed in this study and classified as *Intrusive Control* and *Food availability/accessibility* practices, respectively.

[1] Instruments completed both by parents and children

[2] Instruments only completed by children

## Behavior change techniques used in the intervention

Table 3 lists the BCTs identified in the intervention descriptions available in the selected articles and other published papers about the programs. A detailed definition of the identified BCTs and the corresponding descriptions in the articles (final decision) is available in the S3 Table. We found 13 of the 16 BCTs clusters were used, ranging from 1 to 12, with an average

of 6.45 (SD = 3.43) per intervention. The BCT cluster *Shaping knowledge* was included in all programs; other clusters were also coded in more than 50% of the interventions: *Comparison of behavior*, *Natural consequences*, *Identity*, and *Antecedents*. The BCT categories *Scheduled consequences*, *Self-belief*, and *Covert learning* were not implemented in any intervention. Overall, the programs reported 30 different BCTs of the 96 BCTs classified in Michie et al.'s taxonomy, ranging from 1 to 22, with an average of 9.89 (SD = 6.85) BCTs per intervention. The BCTs *Instruction on how to perform the behavior*, *Demonstration of the behavior*, *Identification of self as role model*, *Information about social and environmental consequences*, and *Restructuring the physical environment* were used in most interventions (> 50%). BCTs like *Commitment*, *Prompts/cues*, or *Behavior substitution* were rarely mentioned (N = 1).

A diversity of BCTs were used in some clusters. *Goals and planning* was implemented through five different BCTs: *Goal setting (behavior)*, *Problem solving*, and *Action planning* across four programs; *Review behavior goal(s)*, and *Commitment* in one of the programs. Each of the two BCT clusters most included in the interventions, *Natural consequences*, and *Antecedents*, included three different BCTs. For five BCT clusters, *Shaping knowledge*, *Associations*, *Reward and threat*, *Regulation*, and *Identity*, only one BCT was found.

## Quantitative synthesis of parental feeding practices as outcomes

When all time points were considered, non-significant effects were identified for all dimensions, except for *Food availability/accessibility*, which benefited the intervention group (Table 4). A narrower yet non-significant effect was found explicitly for healthy food availability/accessibility ($p$ = .080). For T2, effects for all dimensions were not significant. Heterogeneity values ranged from small to higher values, leading to broad 95% confidence intervals and prediction intervals. The study that provided the highest precision in the results (small standard deviation and variance) was the *Early Food for Future Health* program. All prediction intervals suggest future studies may report effects from different magnitudes, ranging from positive to negative or even nonexistent. Forest plots depict the weighted average effects per study, by dimension and time points (Figs 3 and 4).

## Discussion

The current systematic review and meta-analysis studied web-based programs to promote positive feeding practices in parents of children up to 12 years old with published RCTs results. In this work, we addressed three main questions: a) Which parental feeding practices are studied as outcomes of the programs?; b) Which behavior change techniques are used to promote changes in parental feeding practices?; and c) What is the effectiveness of these interventions on changing the different parental feeding practices?.

Twelve studies regarding nine different programs complied with the defined criteria. The samples analyzed in most trials were relatively small, partially due to difficulties in retaining participants, especially from the recruitment to the baseline evaluation [49, 55, 59]. In our review, 7 of the 12 studies (corresponding to 5 interventions) were judged as having a high risk of bias, mainly in the *Randomization process* and *Measurement of the outcome*. Further intervention trials should contribute to more comprehensive trial designs and detailed information about the method of blinding chosen [61, 62].

### Identification of parental feeding practices defined as outcomes in the studies

We found that all trials considered at least one structuring parental feeding practice as an outcome; the *Food availability/accessibility* was the most frequently studied feeding practice in

**Table 3. BCTs and BCT clusters implemented in each program, according to Michie et al.'s taxonomy (2013).**

| | HomeStyles | Family Eats | Feeding Healthy Food to Kids | Project FUN | EMPOWER | Time2bHealthy | Early Food For Future Health | Happier Meals | 5-4-3-2-1-0 Program | *Sum* |
|---|---|---|---|---|---|---|---|---|---|---|
| **1. Goals and planning** | X | X | | | X | X | | | | **4** |
| 1.1. Goal setting (behavior) | x | x | | | x | x | | | | 4 |
| 1.2. Problem solving | x | x | | | x | x | | | | 4 |
| 1.4. Action planning | x | x | | | x | x | | | | 4 |
| 1.5. Review behavior goal(s) | | | | | | x | | | | 1 |
| 1.9. Commitment | | | | | x | | | | | 1 |
| **2. Feedback and monitoring** | X | X | | | X | X | | | | **4** |
| 2.2. Feedback on behavior | | x | | | | x | | | | 2 |
| 2.3. Self-monitoring of behavior | x | x | | | x | x | | | | 4 |
| 2.4. Self-monitoring of outcome(s) of behavior | | x | | | | | | | | 1 |
| **3. Social support** | X | | | X | X | X | | | | **4** |
| 3.1. Social support (unspecified) | x | | | x | x | x | | | | 4 |
| 3.2. Social support (practical) | | | | | x | x | | | | 2 |
| **4. Shaping knowledge** | X | X | X | X | X | X | X | X | X | **9** |
| 4.1. Instruction on how to perform the behavior | x | x | x | x | x | x | x | x | x | 9 |
| **5. Natural consequences** | x | | | X | X | X | X | | X | **6** |
| 5.1. Information about health consequences | x | | | | x | | x | | x | 4 |
| 5.3. Information about social and environmental consequences | x | | | x | x | x | x | | | 5 |
| 5.6. Information about emotional consequences | x | | | | x | | x | | | 3 |
| **6. Comparison of behavior** | X | X | | X | X | X | X | X | X | **8** |
| 6.1. Demonstration of the behavior | x | x | | | x | x | x | x | x | 7 |
| 6.2. Social comparison | | | | x | x | x | | | | 3 |
| **7. Associations** | X | | | | | | | | | **1** |
| 7.1. Prompts/cues | x | | | | | | | | | 1 |
| **8. Repetition and substitution** | | | | X | X | X | | | X | **4** |
| 8.1. Behavioral practice/rehearsal | | | | x | x | | | | x | 3 |
| 8.2. Behavior substitution | | | | | x | | | | | 1 |
| 8.7. Graded tasks | | | | | x | x | | | | 2 |
| **9. Comparison of outcomes** | X | | | | X | X | | | X | **4** |
| 9.1. Credible source | x | | | | | | | | x | 2 |
| 9.2. Pros e cons | x | | | | x | x | | | | 3 |
| **10. Reward and threat** | | | | | X | | | | | **1** |
| 10.4. Social reward | | | | | x | | | | | 1 |
| **11. Regulation** | X | | | | X | | | | | **2** |
| 11.2. Reduce negative emotions | x | | | | x | | | | | 2 |
| **12. Antecedents** | X | X | | | X | | X | X | | **5** |

*(Continued)*

**Table 3.** (Continued)

| | HomeStyles | Family Eats | Feeding Healthy Food to Kids | Project FUN | EMPOWER | Time2bHealthy | Early Food For Future Health | Happier Meals | 5-4-3-2-1-0 Program | *Sum* |
|---|---|---|---|---|---|---|---|---|---|---|
| 12.1. Restructuring the physical environment | x | x | | | x | | x | x | | 5 |
| 12.2. Restructuring the social environment | x | | | | x | | x | x | | 4 |
| 12.5. Adding objects to the environment | x | | | | | | | | | 1 |
| **13. Identity** | X | X | | X | X | | X | X | | 6 |
| 13.1. Identification of self as role model | x | x | | x | x | | x | x | | 6 |
| *Sum (of the categories of BCT used)* | 11 | 6 | 1 | 6 | 12 | 8 | 5 | 4 | 5 | 58 |
| *Sum (of the specific BCT used)* | 18 | 10 | 1 | 6 | 22 | 14 | 8 | 5 | 5 | 89 |

this category. Children's food environment undergoes significant changes when parents start to provide more healthy foods and control the access to unhealthy foods; thus, those changes can directly influence children's eating behavior and facilitate the adoption of other positive

**Table 4. Hedges g, heterogeneity, and prediction intervals for the pooled studies and nested effects (N = 8, k = 68).**

| | All Available Time Points Included | | | | T2 | | | |
|---|---|---|---|---|---|---|---|---|
| **Dimensions** | *g*(SE) | 95% CI | I$^2$ (%) | 95% Prediction Interval | *g*(SE) | 95% CI | I$^2$ (%) | 95% Prediction Interval |
| **Restriction** | -.05(.12) | [-.42, .33] | 39,7 | [-.62, .53] | .02(.09) | [-.43, .46] | 0 | [-.43, .46] |
| (*n* = 5, *k* = 8; *n* = 3, *k* = 3) | | | | | | | | |
| **Pressure to eat** | -0.23(.17) | [-.69, .24] | 64,4 | [-1.37, .91] | -.32(.29) | [-1.58, .94] | 64,3 | [-2.44, 1.80] |
| (*n* = 5, *k* = 8; *n* = 3; *k* = 3) | | | | | | | | |
| **Food to control negative emotions** | -.08(.19) | [-2.45, 2.30] | 43,3 | [-3.51, 3.36] | -.08(.19) | [-2.45, 2.30] | 43,3 | [-3.51, 3.56] |
| (*n* = 2, *k* = 2; *n* = 2, *k* = 2) | | | | | | | | |
| **Monitoring** | -.12(.30) | [-1.43, 1.18] | 75,3 | [2.57, 2.33] | .17(.23) | [-2.77, 3.10] | 0,41 | [-2.77, 3.10] |
| (*n* = 3, *k* = 5; *n* = 2, *k* = 2) | | | | | | | | |
| **Meal and snack routines** | -.01(.04) | [-.33, .32] | 24,9 | [-.54, .53] | 0(.04) | [-.32, .32] | 26,5 | [-.54, .54] |
| (*n* = 3, *k* = 11; *n* = 3, *k* = 10) | | | | | | | | |
| - Positive | .01(.03) | [-.33, .34] | 80,9 | [-3.28, 3.29] | .01(.03) | [-.33, .34] | 80,9 | [-3.28, 3.29] |
| (*n* = 2, *k* = 4; n = 2, k = 4) | | | | | | | | |
| - Negative | -.08(.06) | [-.84, .68] | 0 | [-1.56, 1.40] | -.08(.08) | [-1.05, .90] | 0 | [-1.05, .90] |
| (*n* = 2, *k* = 2; *n* = 2, *k* = 2) | | | | | | | | |
| **Modeling** | -4.16(5.91) | [-45.80, 37.50] | 98,8 | [-47.10, 38.78] | -3.99(4.62) | [-24.10, 16.10] | 98,8 | [-26.50, 18.52] |
| (*n* = 3, *k* = 5; *n* = 3, *k* = 3) | | | | | | | | |
| **Food availability/accessibility** | .46(.14) | [.08, .85] | 97,1 | [-4.07, 5.00] | .93(.50) | [-.46, 2.32] | 97,1 | [-21.11, 21.97] |
| (*n* = 5, *k* = 42; *n* = 5, *k* = 16) | | | | | | | | |
| - Healthy | .27(.08) | [-.08, .60] | 98,5 | [-2.97, 3.50] | 1.22(.77) | [-2.10, 4.55] | 98,5 | [-9.26, 11.70] |
| (*n* = 3, *k* = 29; *n* = 3, *k* = 15) | | | | | | | | |
| - Unhealthy | .98(.82) | [-2.54, 4.50] | 97,8 | [-3.46, 5.43] | 1.46(1.31) | [-4.18, 7.1] | 98,1 | [-9.79, 12.71] |
| (*n* = 3, *k* = 12; *n* = 3, *k* = 7) | | | | | | | | |
| **Food preparation** | -2.1(2.43) | [-33.00, 28.80] | 99,5 | [-63.47, 59.27] | -7.34(7.71) | [-105.00, 90.60] | 99 | [-129.69, 115.01] |
| (*n* = 2, *k* = 13; *n* = 2, *k* = 7) | | | | | | | | |
| **Encourage healthy eating** | .05(.04) | [-.49, .59] | 0 | [-.58, .59] | .06(.07) | [-.82, .94] | 0 | [-.82, .94] |
| (*n* = 2, *k* = 3; *n* = 2, *k* = 2) | | | | | | | | |

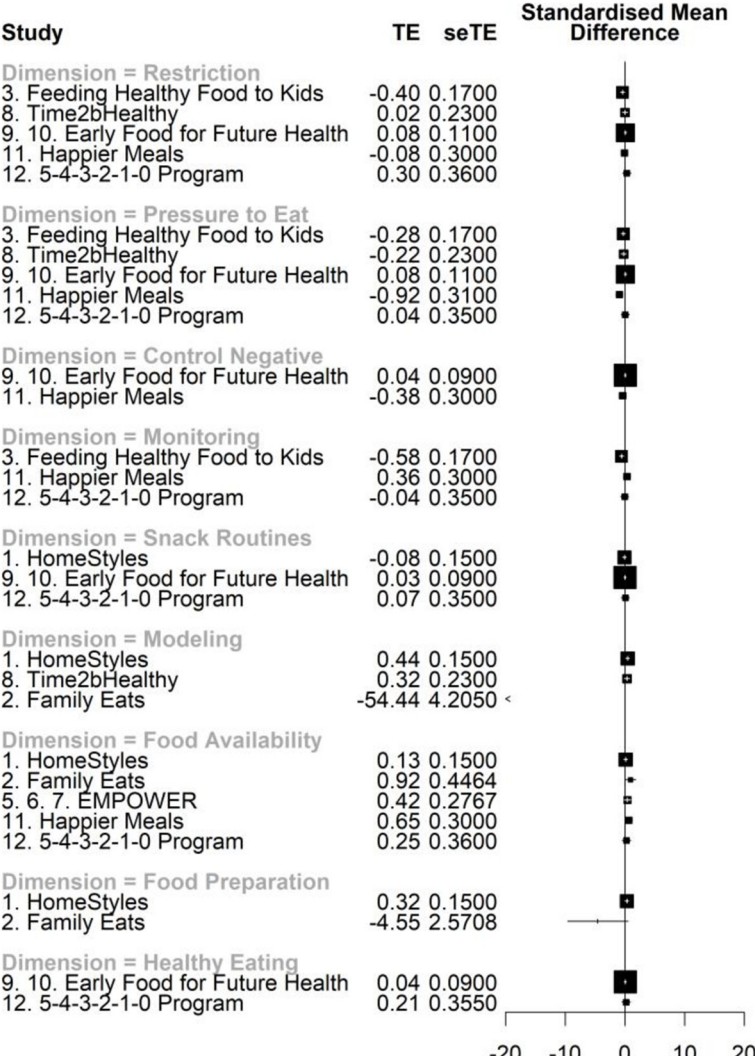

**Fig 3. Forest plot of the effects of web-based interventions on parental feeding practices, estimated for all time points.** Legend: TE = treatment effect, and seTE = standard error treatment effect. About values on standardized mean differences: zero values indicate lack of effect, negative values favor the intervention group, and positive values favor the control group. Information about the outlier (*Family Eats*) for *Modeling* dimension: Family Eats: *g* = -54.4450, SE = 4.21, 95% CI [-62.69; -46.20].

parental feeding practices, such as guided choices [8]. Most trials assessed *Pressure to eat* and *Restriction*, two coercive controlling practices that have shown clear associations with children's unhealthy eating behaviors [11]. Nevertheless, few programs included outcomes regarding parental feeding practices to promote children's autonomy regarding eating, even when those positive practices appeared to be promoted in some interventions [52, 57, 59]. Specifically, only two programs [49, 51] measured parent's behaviors to promote children's self-regulation, e.g., through the identification of hunger and satiety cues (identified as the *Encourage healthy eating* practice in O'Connor et al., 2017). Although the child's mechanisms of food intake self-regulation are innate [63], they change due to the influence of the environment; parents' behaviors can contribute significantly to this process [64, 65]. Beckers et al. (2021) recently concluded that autonomy supporting practices and some structuring practices (e.g.,

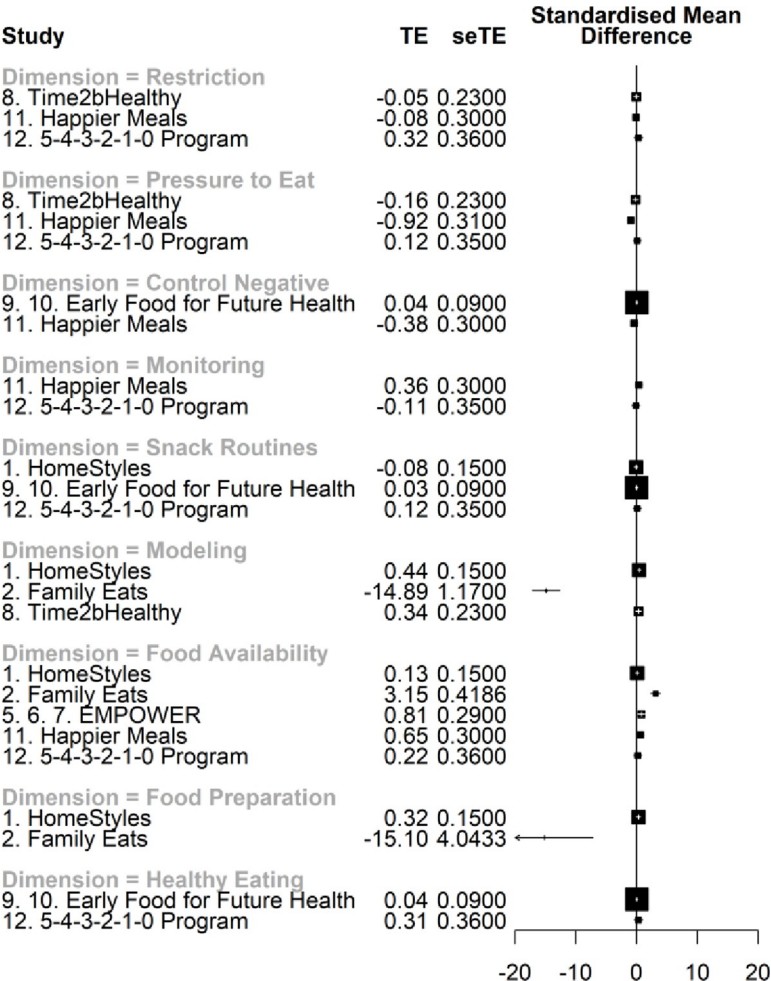

**Fig 4. Forest plot of the effects of web-based interventions on parental feeding practices, estimated at T2.** Legend: TE = treatment effect, and seTE = standard error treatment effect. About values on standardized mean differences: zero values indicate lack of effect, negative values favor the intervention group, and positive values favor the control group. Information about the outlier (*Family Eats*) for *Food Preparation* dimension: Family Eats: $g$ = -15.10, SE = 4.04, 95% CI [-23.02, -7.18].

rules and limits) were understudied and inconsistently measured in prospective quantitative studies with parents of 2–18 years old children [66]. The adoption of positive practices that provide a flexible but structured home food environment, where children can participate and be involved in developing their healthy eating behaviors, is possible at all developmental stages and allows a positive parental continuum of influence since the child's early years [13]. Researchers should continue to invest in including child's autonomy promoting feeding practices as outcome measures in interventions.

## Identification of BCTs implemented in the interventions

The interventions varied greatly regarding the diversity and quantity of BCTs used. The programs invested substantially in techniques to teach how to apply a particular feeding practice or perform specific behaviors related to food choices and preparation (*Instruction on how to perform the behavior*). The provision of explanations about a specific behavior is a traditional behavioral methodology in health interventions and was identified in earlier systematic

reviews of programs to promote children's healthy diet [31, 32, 67]. Consistently with the SCT model endorsed by most programs, *Demonstration of the behavior* was frequently implemented in the interventions. Parents watched videos or vignettes, showing parents' feeding practices during mealtime, or listened to first-person reports of feeding challenges, decisions, and the consequences of specific behaviors. This technique was frequently complemented with the message that parents should be good role models for the child and that young children often follow their parents' eating behaviors (*Identification of self as role model*), which might facilitate parents' adherence to change. Technological resources may facilitate the demonstration of specific behaviors through animations, cartoons, and videos. Also, BCTs regarding *Natural consequences* and *Antecedents* were used in several programs. In addition to a general explanation of the relationship between healthy eating and children's health, some studies included information on how adopting certain parental feeding practices can change the children's eating context and behavior and the quality of the parent-child interactions at mealtimes. Advice to restructure the child and family's physical or social eating environment might be related to a significant emphasis on promoting structuring feeding practices, as discussed earlier.

BCTs regarding *Goals and planning* and *Feedback and monitoring*, although central to the SCT model, were used less often. Half of the programs using self-monitoring did not provide any feedback about parents' progress. Similarly, only one of the four programs that used goal-setting reviewed the behavior goals with parents and adjusted them according to their achievement. Self-regulatory strategies play a critical role in engaging individuals in health behavior changes [68, 69]. Healthy eating and physical activity interventions that combine self-monitoring with at least one other self-regulatory strategy proved to be more effective than programs that did not include those techniques [70]. The percentage of interventions that applied these strategies did not differ significantly from other reviews about general online interventions on nutrition behaviors [14] or to change general health behaviors [71]. However, *Goal setting* was the BCT most used in individual or group interventions to support parents in reducing children's unhealthy food intake [31]. While the monitoring of parental practices, the definition of specific goals, and parents' report of behavior changes can be implemented autonomously with minimal guidance, the provision of individualized feedback might be more demanding. Technology based interventions allow tailored, immediate and standardized feedback according to parent's responses, through algorithms and decision rules that program the system to provide specific feedback (descriptive, evaluative, comparative) according to the participants' responses or performance [72]. Nevertheless, tailored feedback implies parents' detailed self-monitoring regarding their child feeding behaviors. Researchers must decide on the right balance between collecting enough information to allow for tailoring and not burdening parents with excessive data collection and records.

Only two interventions (*HomeStyles* and *EMPOWER*) sought to advise parents on reducing their negative emotions in managing children's feeding challenges. These actions can be relevant to the promotion of positive parental feeding practices. Parental negative emotional responses (e.g., frustration, disappointment, concern) to children's refusal of vegetables were a predictor of non-maintenance of regular positive communications about vegetables [73]. Also, parents' distress in response to the child's negative emotions predicted non-responsive feeding practices in mothers with binge eating problems [74].

## Efficacy of interventions regarding parental feeding practices: results from the meta-analysis

The forest plots indicate that most programs' effects on parental practices are minimal, almost always on top of the centerline (zero representing no effects). These results may be due to the

reduced number of studies retained and, consequently, to the limited data available. It is also noteworthy that many trials included a control condition with some sort of intervention, offering, in some cases, similar contents to the active treatment. Although this option can help differentiate the effect of the active ingredient and preserve the internal validity and blinding features of the trials, it can unintentionally impact the outcomes and reduce the differences between experimental and control groups [75]. The type of samples collected should also be considered. All programs aimed to promote healthy children's eating habits and/or prevent childhood obesity in community samples. Therefore, it is expectable that most parents included in the studies did not use extremely inappropriate parental feeding practices and that their children had overall healthy eating behaviors. Additionally, parents more motivated to enroll in these interventions may not represent the most problematic populations. The study of the efficacy of these web-based interventions with more heterogeneous samples is suggested.

*Food availability/accessibility* was the only parental feeding practice for which the effects were significant, benefiting the intervention group when all time points were considered. All these programs aimed to promote a healthy food environment. Although the interventions differed regarding their duration and dosage, they all included contents and resources on improving the availability of healthy foods at home through planning grocery purchases and menus, choosing healthy foods, involving the child in meal preparation, and adapting cooking recipes. Parents may find it easier to change their home food environment and the kind of foods they make available/accessible to their children, compared with other feeding practices that imply direct interaction with the child (e.g., rules and limits, pressure to eat) or changes in their own eating behavior (e.g., modeling). Nevertheless, the positive changes on this particular feeding practice may take longer to consolidate, as suggested by the results of our study for all time points' measurements.

## Limitations and future directions

Several limitations should be considered. Although several web-based programs aimed at improving children's dietary patterns that involve parents are available, few studies defined parental feeding practices as an outcome. The studies were quite heterogeneous regarding data collection (i.e., diversity of parental behaviors studied, use of different measures), their samples (e.g., child age, socio-economic status), and intervention components (e.g., number of BCTs, more or less focus on child feeding). The small number of studies did not allow conducting all meta-analytical procedures, such as meta-regression analyses, a sensitivity analysis for publication bias, or a moderator analysis, which could help clarify influences on outcomes. Also, the sample size made the exclusion of studies with a high risk of bias impracticable. Finally, despite estimating the effects using all information available, which increased the number of nested effects within studies, results should be cautiously interpreted and considered a preliminary look at the parental feeding practices outcomes provided by these programs. We considered only articles written in English, excluded the grey literature, and only analyzed published data. Although we recognize that these criteria might have limited access to other works, we believe we identified the most relevant literature on this topic since the search was run in four relevant databases in this field without time restriction.

## Conclusions

Although promising, parental web-based interventions that promote children's healthy eating habits and assess changes on parental feeding practices are still scarce, and several publications about those interventions had a high risk of bias. With the available data, we found that most

programs' effects were small and non-significant for the evaluated parental practices, except for *Food availability/accessibility* when considering all follow-up measurements. The development of high-quality controlled trials with larger samples and detailed reports is needed to determine with greater certainty the impact of the intervention on parental feeding practices. More frequent use of measures to evaluate parental support of children's autonomy in the feeding context might be considered, depending on the intervention's objectives and the child's developmental stage. Although we recognize the effort to design interventions based on a theoretical framework and including various BCTs, more regular inclusion of self-regulatory strategies could be relevant, taking advantage of technological resources.

## Supporting information

**S1 File. PRISMA checklist.**
(DOC)

**S2 File. Risk of bias assessment (RoB2 tool).**
(XLSM)

**S1 Table. Summary of parental web-based interventions that assessed parental feeding practices as outcome.** Legend: CG = control group and IG = intervention group.
(DOCX)

**S2 Table. Categorization of parental feeding practices assessed on studies (conceptual framework proposed by O'Connor et al., 2017, and Vaughn et al., 2016).**
(XLSX)

**S3 Table. Behavior change techniques and clusters identified in each intervention (BCTTv1 taxonomy, Michie et al., 2013).**
(XLSX)

## Author Contributions

**Conceptualization:** Ana Isabel Gomes, Ana Isabel Pereira, Luisa Barros.

**Data curation:** Ana Isabel Gomes, Magda Sofia Roberto, Klara Boraska, Luisa Barros.

**Formal analysis:** Ana Isabel Gomes, Ana Isabel Pereira, Magda Sofia Roberto, Klara Boraska, Luisa Barros.

**Funding acquisition:** Luisa Barros.

**Investigation:** Ana Isabel Gomes, Ana Isabel Pereira, Magda Sofia Roberto, Klara Boraska, Luisa Barros.

**Methodology:** Ana Isabel Gomes, Ana Isabel Pereira, Magda Sofia Roberto, Luisa Barros.

**Project administration:** Ana Isabel Pereira, Luisa Barros.

**Resources:** Ana Isabel Gomes, Magda Sofia Roberto.

**Software:** Magda Sofia Roberto.

**Supervision:** Ana Isabel Pereira, Luisa Barros.

**Validation:** Ana Isabel Pereira, Luisa Barros.

**Visualization:** Ana Isabel Gomes, Magda Sofia Roberto, Klara Boraska.

**Writing – original draft:** Ana Isabel Gomes, Ana Isabel Pereira, Magda Sofia Roberto, Klara Boraska, Luisa Barros.

**Writing – review & editing:** Ana Isabel Gomes, Ana Isabel Pereira, Magda Sofia Roberto, Klara Boraska, Luisa Barros.

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
