## [Decision Letter · Decision Letter 0]

23 Feb 2021

PONE-D-21-02383

Changing parental feeding practices through web-based interventions: A systematic review and meta-analysis

PLOS ONE

Dear Dr. Gomes,

Thank you for submitting your manuscript to PLOS ONE. After careful consideration, we feel that it has merit but does not fully meet PLOS ONE’s publication criteria as it currently stands. Therefore, we invite you to submit a revised version of the manuscript that addresses the points raised during the review process.

In addition to the issues identified by the reviewers, please address the minor writing/ grammatical issues that I have identified in my review, which are listed at the end of this letter.

We look forward to receiving your revised manuscript.

Kind regards,

Jane Anne Scott, PhD, MPH Grad Dip Dietetics, BSc

Academic Editor

PLOS ONE

Journal Requirements:

2. Please amend the manuscript submission data (via Edit Submission) to include author Ana Isabel Gomes.

3. Please amend your authorship list in your manuscript file to include author Fernandes Fernandes Gomes.

Additional Editor Comments:

Please address these minor grammatical/ writing issues:

Line 62 remove duplicate words ‘actions to’

Line 67 and elsewhere I think that this should read ‘structuring’ feeding practices (as used in line525).

Line 92 for clarity and consistency I suggest you say ‘specific parental feeding practices’.

Line 124 should read ‘a) it targeted…’

Lines 196-197 this sentence is awkwardly worded. I am not sure what is meant by “intervention condition’s actions”. Is the word condition needed here?

Line 278 should read ‘focused on other topics’.

Lines 281 and 282 rather than say ‘most studies’ please specify the number of studies, as you have done for other characteristics.

Lines 444-445 This sentence needs to be reworded as it is ambiguous and suggests that the interaction between participant and program staff is only one way, i.e. the parents contacting staff. Presumably staff were contacting parents to encourage participation or provided feedback about goals’ accomplishment.

Lines 503-505 This sentence only identifies the direction of the difference in the number of BCTs identified in this review and the systematic reviews but does not provide any indication of the magnitude of the difference. It would be useful, if possible, to include the magnitude of difference by comparing the actual mean numbers of BCTs.

Lines 551 to 552 This sentence seems incomplete. What is meant by’ mothers with binge’. Do you mean mothers with binge eating problems or who binge eat?

Line 573 should read ‘benefiting the intervention group’

Line 582 suggest rewording ‘a near significant effect’ or an ‘effect nearing significance’.

Line 594 awkward wording, suggest rewording ‘…. did not allow the performance of all.

Line 598 suggest rewording ‘… should be interpreted cautiously and consider to be a preliminary look…’

Reviewers' comments:

Reviewer's Responses to Questions

**Comments to the Author**

1. Is the manuscript technically sound, and do the data support the conclusions?

Reviewer #1: Yes

Reviewer #2: Partly

2. Has the statistical analysis been performed appropriately and rigorously? 

Reviewer #1: Yes

Reviewer #2: Yes

3. Have the authors made all data underlying the findings in their manuscript fully available?

Reviewer #1: Yes

Reviewer #2: Yes

4. Is the manuscript presented in an intelligible fashion and written in standard English?

Reviewer #1: Yes

Reviewer #2: Yes

5. Review Comments to the Author

Reviewer #1: This manuscript describes a systematic review and meta-analysis of web-based interventions targeting the feeding practices of parents of children aged up to 12 years, for improving dietary intake and the prevention of obesity. The review found that there were few studies addressing autonomy and support-based strategies, and that food availability and accessibility were the only practices which were significantly influenced by web-based interventions. Unlike traditional interventions in this space, the BCT’s goals and planning, and feedback and monitoring were used less frequently than those related to shaping knowledge. This manuscript provides a comprehensive and sound critique of the available literature, using appropriate methods, thus contributing important knowledge to this field. The focus on web-based interventions and the inclusion of behaviour change technique coding of interventions makes the work topical and valuable.

This review and meta-analysis was clearly a substantial undertaking, and the authors should be congratulated for their efforts. However, my main concern is the length and clarity of the manuscript, with the discussion section alone being around 3000 words. This is excessive, even for a systematic review, and is detrimental to the overall findings and key ‘take-home’ messages of the work. I suggest that this article should be substantially reduced in length, condensing some results into summary tables (e.g. such as study characteristics) and focusing on the main findings as they relate to the primary objectives of the work.

Specific feedback:

– Page 5, line 101 - ‘Behavior’ change taxonomy / techniques, rather than ‘behavioral’ throughout

- Page 10, line 186 - Did only one author code BCT content? Please specify if this was the case.

– Page 11, line 97 - How did you identify BCT's related to the nutritional component of the study only? In my experience this is not always clear and can be difficult to identify from published descriptions of interventions. Did you consider coding unpublished content/intervention materials (e.g. websites where available)?

– Page 13, line 237 - For clarity please use consistent terminology to define the articles / publications versus programs / interventions / studies.

– The risk of bias (page 13 & 14) and study/sample characteristics (pages 14-16) sections of the results should be reduced. This could be summarised in a table, with the text only highlighting the most important findings (this should also be applied to the remainder of the results section).

– The first 3.5 pages of the discussion (pages 24-27) relate to aspects of study design, which although important, are superfluous to the primary objectives of the review, therefore could be reduced substantially. Also consider rewording to discuss your main findings first.

- Page 31, line 548 - please remind the reader of what you are referring to when using the term 'negative emotions' (i.e. pressure to eat)

- Page 32, line 572 - I would like to see some further discussion (and key references) around why food availability and accessibility might be the only parental feeding practice with significant effects (i.e. easier to change / at a higher socioecological level??), and why this might be important / what does this mean for future research?

- Page 31, line 552 - incomplete sentence?

– Reduce the number of references – there were only 12 articles included in the review, however you have 100 references which is excessive and unnecessary.

– Can the scale on the forest plots be modified? At present the scale is large due to a single outlier. It is therefore difficult to see the other results.

Reviewer #2: Manuscript ID: PONE-D-21-02383

Title: Changing parental feeding practices through web-based interventions: A systematic

review and meta-analysis

Thank you for the opportunity offered to review this paper. Systematic reviews and meta-analyses are very important to enhance knowledge on specific topics. This article has the potential to fill in an important lack in the literature. I have several comments meant to stimulate the authors to further improve this piece.

General comments:

1. The Discussion section can be improved substantially I guess (mainly the first part, e.g., page 25 and 26). That part includes separate paragraphs that do not follow each other very clearly and that often end with conclusions based on other studies. Please focus on the main implications following from your own study findings. More important, some research questions do not seem to be correctly framed. In the Introduction and Discussion section the following was stated: “Which parental feeding practices do the interventions aim to promote or modify?”, while the results and Table 2 present the categorization of parental feeding practices considered as outcome measures, and thus whether practices were evaluated per intervention. Targeting or addressing certain food parenting practices in interventions is not similar to reporting about parenting practices as outcome measures. Please explain. And make sure that research questions are clearly and consistently presented, aligning with the subsequent information provided.

2. Moreover, specific framing can be improved at some points. Please avoid too long sentences throughout your piece (e.g., page 11; lines 88-91: Interventions with preschool children that included repeated taste exposure and reward had better results regarding vegetable intake compared with those which did not [22], and, for children under three, exposure (with or without consumption) to various and unfamiliar vegetables also diminished the reluctance to try a new vegetable [23].

Some further specifications:

3. Abstract: “The development of high-quality and controlled trials with larger samples are needed to determine with greater certainty the impact of the intervention on parental feeding

behaviors.” = unclear what is meant by ‘the intervention’. Please specify.

4. Page 3; Line 45-46: “Adults are recognized as a major influence on children’s dietary patterns and can shape their preferences and food acceptance, guiding children’s eating behaviors during mealtimes, as positive role models, and through early exposure to different healthy foods [3, 4].” Very long and bit unclear sentence. Please adapt.

5. Page 3; line 62: typo ‘parent’s actions to actions’. Please adjust.

6. Page 4; line 68: correlations are reported here. Please reframe this part and clearly report about latest insight based on longitudinal and experimental data.

7. Page 5; line 86-87 “For children up to five years, interventions that target child feeding practices (e.g., repeated exposure, flavor-flavor learning, prompting, modeling) effectively increase fruit and vegetable intake and lasting up to 12 months [21].”: sentence unclear, please change for instance into ‘, with effects lasting…’

8. The term ‘structure parental feeding practices’ is used throughout this piece, however, I am wondering whether this is a correct term, and whether ‘structure-related parental feeding practices’ would be a better term.

9. Page 14; line 151-152: “All the steps of the studies’ selection were conducted independently by two authors according to the defined inclusion/exclusion criteria.” What were these steps? This becomes clear from the flow diagram, but it may be helpful to shortly mention in the text that a two-phase procedure has been used in which first titles/abstracts and then full items were screened. And please mention for all these kind of sentences: “In the case of rating conflicts, both authors discussed and reached a consensus, helped by a third author” who these authors were by indicating their initials.

10. Page 21; line 278: “Five programs also focused other topics” = also focused on other topics?

11. Were results similar found when studies having a high risk of bias were excluded? Mainly important for the last research question and meta-analysis.

12. General BCT use within the interventions has been reported. However, it might be more interesting to link BCT use to specific food parenting practices. This would also better align with the reported aim: line 405: “second, to identify behavioral change strategies that are chosen to change those parental practices”.

13. Page 24; line 406: “and third, to analyze the impact of the interventions on those parental outcomes.” Reference to ‘those parental outcomes’ is a bit unclear. Please specify.

14. Page 24; line 406-407: “Overall, we found that i) coercive controlling and structure parental feeding practices are commonly assessed”. Please also report in this main summary of findings which food parenting practices were not commonly assessed.

15. Page 25; line 414: “This small number of articles highlights the difficulty of finding programs assessing…”. I think this conclusion is somewhat weird. This probably highlights the fact that only few such intervention programs focused on the assessment of FPPs.

16. Page 29; line 499-500: “Future programs should invest in raising parents' awareness of the use of this child-centered practices, and continue to work on the validation of instruments that appraise those specific parental behaviors. Recent intervention studies have often targeted responsive feeding. Please focus on implications of your main findings, that is, on feeding practices as outcome measures in interventions if I am correct.

17. Page 29: line 505-506: “Two programs with more BCTs (HomeStyles and EMPOWER) were described in several published articles about the program, which may have increased BCTs' global mean.” How is this possible? Given that the following procedure was reported on page 12, line 215-216 “Because different publications reported on the results from the same program, data were analyzed considering programs.

18. Page 25; line 412: “except for Food availability and accessibility when considering all follow-up measurements”. What does this mean? Please explain.

19. Page 32; line 582: “In this review, we found an effect near significance on the availability of healthy food. These results suggest that changing children’s food environment regarding unhealthy foods is more challenging to achieve than increasing the availability of healthy foods.” Please further explain, because no effects were found on changing children’s food environment regarding healthy foods?

6. PLOS authors have the option to publish the peer review history of their article (what does this mean?). If published, this will include your full peer review and any attached files.

Reviewer #1: **Yes: **Chelsea Emma Mauch

Reviewer #2: **Yes: **Junilla Larsen

---

## [Author Response · Author response to Decision Letter 0]

10 Mar 2021

Comments from the editor:

 2. Please amend the manuscript submission data (via Edit Submission) to include author Ana Isabel Gomes.

 3. Please amend your authorship list in your manuscript file to include author Fernandes Fernandes Gomes.

Answer: Thank you for raising this issue. I had already identified this issue in the first submission of the manuscript, but I did not know why this happened did not find a way to solve it. Both profiles are related to one author (Ana Isabel Fernandes Gomes). I hope that in this submission the problem is solved.

Additional Editor Comments

Please address these minor grammatical/ writing issues:

- Line 62 remove duplicate words ‘actions to’:

Corrected.

- Line 67 and elsewhere I think that this should read ‘structuring’ feeding practices (as used in line525):

 Thank you for your comment. We adopted your suggestion and changed all “structure feeding practices” to “structuring feeding practices”.

- Line 92 for clarity and consistency I suggest you say ‘specific parental feeding practices’.:

We have corrected this sentence.

- Line 124 should read ‘a) it targeted…’:

We have corrected this sentence.

- Lines 196-197 this sentence is awkwardly worded. I am not sure what is meant by “intervention condition’s actions”. Is the word condition needed here?:

We deleted the word “condition”.

- Line 278 should read ‘focused on other topics’:

We have corrected this sentence.

- Lines 281 and 282 rather than say ‘most studies’ please specify the number of studies, as you have done for other characteristics.:

We have completed this sentence according to your suggestion: “Six interventions targeted parents of 2 to 6 years old children exclusively.”

- Lines 444-445 This sentence needs to be reworded as it is ambiguous and suggests that the interaction between participant and program staff is only one way, i.e. the parents contacting staff. Presumably staff were contacting parents to encourage participation or provided feedback about goals’ accomplishment.:

We have completed this sentence according to your suggestion: “Although the programs were exclusively web-based, five promoted the interaction between participants and staff, to clarify doubts…”.

- Lines 503-505 This sentence only identifies the direction of the difference in the number of BCTs identified in this review and the systematic reviews but does not provide any indication of the magnitude of the difference. It would be useful, if possible, to include the magnitude of difference by comparing the actual mean numbers of BCTs.:

Thank you for the suggestion. Considering the need to reduce substantially the discussion, this sentence was deleted.

- Lines 551 to 552 This sentence seems incomplete. What is meant by’ mothers with binge’. Do you mean mothers with binge eating problems or who binge eat?:

Thank you for pointing this, we completed this sentence: “Also, parents' distress in response to the child’s negative emotions predicted non-responsive feeding practices in mothers with binge eating problems.”

- Line 573 should read ‘benefiting the intervention group’:

We have corrected this sentence.

- Line 582 suggest rewording ‘a near significant effect’ or an ‘effect nearing significance’.:

We have corrected this sentence, following your second option.

- Line 594 awkward wording, suggest rewording ‘…. did not allow the performance of all.:

We have corrected this sentence.

- Line 598 suggest rewording ‘… should be interpreted cautiously and consider to be a preliminary look…’: 

We have corrected this sentence.

Comments from Reviewer #1:

This manuscript describes a systematic review and meta-analysis of web-based interventions targeting the feeding practices of parents of children aged up to 12 years, for improving dietary intake and the prevention of obesity. The review found that there were few studies addressing autonomy and support-based strategies, and that food availability and accessibility were the only practices which were significantly influenced by web-based interventions. Unlike traditional interventions in this space, the BCT’s goals and planning, and feedback and monitoring were used less frequently than those related to shaping knowledge. This manuscript provides a comprehensive and sound critique of the available literature, using appropriate methods, thus contributing important knowledge to this field. The focus on web-based interventions and the inclusion of behaviour change technique coding of interventions makes the work topical and valuable.

This review and meta-analysis was clearly a substantial undertaking, and the authors should be congratulated for their efforts. However, my main concern is the length and clarity of the manuscript, with the discussion section alone being around 3000 words. This is excessive, even for a systematic review, and is detrimental to the overall findings and key ‘take-home’ messages of the work. I suggest that this article should be substantially reduced in length, condensing some results into summary tables (e.g. such as study characteristics) and focusing on the main findings as they relate to the primary objectives of the work.

– Page 5, line 101 - ‘Behavior’ change taxonomy / techniques, rather than ‘behavioral’ throughout:

Thank you for your suggestion, this was a mistake. We have corrected this word throughout the manuscript.

- Page 10, line 186 - Did only one author code BCT content? Please specify if this was the case.:

Thank you for pointing this, the information was missing. We clarify the categorization process regarding feeding practices and BCTs, and added this sentence: “Both categorizations were performed independently by two authors (A.I.G. and L.B.) and discussed with a third author (A.I.P.) before a final decision was made.“

- Specific feedback:– Page 11, line 97 - How did you identify BCT's related to the nutritional component of the study only? In my experience this is not always clear and can be difficult to identify from published descriptions of interventions. Did you consider coding unpublished content/intervention materials (e.g. websites where available)?:~

The BCT analysis was only applied to the content of the intervention related to children's nutrition or feeding. This analysis was possible because, in programs that included other target behaviors (such as sleep or physical activity), the sessions for each topic were well defined and with detailed explanations of the components of each session. In the analysis, we only considered published articles: articles on program effectiveness, study protocols, or doctoral theses.

– Page 13, line 237 - For clarity please use consistent terminology to define the articles / publications versus programs / interventions / studies.:

Thank you for your suggestion. We revised all the manuscript to use consistent terminology, however, we maintained the same terms: articles / studies / publications vs. programs / interventions. We consider that each article is a study of the program, and, in some cases, we have two or three studies (articles) about the same program.

– The risk of bias (page 13 & 14) and study/sample characteristics (pages 14-16) sections of the results should be reduced. This could be summarised in a table, with the text only highlighting the most important findings (this should also be applied to the remainder of the results section).:

We understand the suggestion. However, descriptive results of the articles and the risk of bias analysis are essential parts of systematic reviews, and we think these should be described in detail. We have provided a table with a summary of the RoB2 results in the supplementary material. As such, we have not made any structural changes to this part of the text.

– The first 3.5 pages of the discussion (pages 24-27) relate to aspects of study design, which although important, are superfluous to the primary objectives of the review, therefore could be reduced substantially. Also consider rewording to discuss your main findings first.:

We recognize that the first part of the discussion is extensive and we made efforts to reduce it, focusing on the main results about studies' characteristics. Considering the limitations of the meta-analysis, we found it more appropriate and helpful to provide a summary of the results according to the research questions.

- Page 31, line 548 - please remind the reader of what you are referring to when using the term 'negative emotions' (i.e. pressure to eat):

We agree, the sentence was not clear. We changed to: “Parental negative emotional responses (e.g., frustration, disappointment, concern) to children’s refusal of vegetables was a predictor of non-maintenance of regular positive communications about vegetables.”

- Page 32, line 572 - I would like to see some further discussion (and key references) around why food availability and accessibility might be the only parental feeding practice with significant effects (i.e. easier to change / at a higher socioecological level??), and why this might be important / what does this mean for future research?:

We reformulated the paragraph to further discuss those results about food availability/accessibility.

- Page 31, line 552 - incomplete sentence?:

Thank you for pointing this, we completed this sentence: “Also, parents' distress in response to the child’s negative emotions predicted non-responsive feeding practices in mothers with binge eating problems.”

– Reduce the number of references – there were only 12 articles included in the review, however you have 100 references which is excessive and unnecessary.:

We understand your concern so we have revised the references. Some references about additional articles consulted for BCTs’ categorization purposed were only mentioned in the supplementary material. Other references were deleted as a consequence of changes in the discussion.

- Can the scale on the forest plots be modified? At present the scale is large due to a single outlier. It is therefore difficult to see the other results.:

We agree with your suggestion so we changed the graphic features and reduced the scale. The outliers in both figures (reporting T2 and all follow-up measurements) are now mentioned in the legend. 

Comments from Reviewer #2:

Thank you for the opportunity offered to review this paper. Systematic reviews and meta-analyses are very important to enhance knowledge on specific topics. This article has the potential to fill in an important lack in the literature. I have several comments meant to stimulate the authors to further improve this piece.

1. The Discussion section can be improved substantially I guess (mainly the first part, e.g., page 25 and 26). That part includes separate paragraphs that do not follow each other very clearly and that often end with conclusions based on other studies. Please focus on the main implications following from your own study findings. More important, some research questions do not seem to be correctly framed. In the Introduction and Discussion section the following was stated: “Which parental feeding practices do the interventions aim to promote or modify?”, while the results and Table 2 present the categorization of parental feeding practices considered as outcome measures, and thus whether practices were evaluated per intervention. Targeting or addressing certain food parenting practices in interventions is not similar to reporting about parenting practices as outcome measures. Please explain. And make sure that research questions are clearly and consistently presented, aligning with the subsequent information provided.:

We agree with your suggestion and reformulated the paragraph to summarize the results regarding the studies’ characteristics. We also clarified what we aimed to study in the research questions, and applied the changes throughout the manuscript: “a) Which parental feeding practices are studied as outcomes of the programs?; b) Which behavior change techniques are used to promote changes in parental feeding practices?; c) What is the effectiveness of these interventions on changing the different parental feeding practices?”

2. Moreover, specific framing can be improved at some points. Please avoid too long sentences throughout your piece (e.g., page 11; lines 88-91: Interventions with preschool children that included repeated taste exposure and reward had better results regarding vegetable intake compared with those which did not [22], and, for children under three, exposure (with or without consumption) to various and unfamiliar vegetables also diminished the reluctance to try a new vegetable [23].:

We revised the manuscript and changed several sentences according to your suggestion. Regarding the sentence mentioned here, we changed to: “Interventions with preschool children that included repeated taste exposure and reward had better results regarding vegetable intake than those that did not [22]. For children under three, exposure (with or without consumption) to various and unfamiliar vegetables also diminished the reluctance to try a new vegetable [23]..”

3. Abstract: “The development of high-quality and controlled trials with larger samples are needed to determine with greater certainty the impact of the intervention on parental feeding

behaviors.” = unclear what is meant by ‘the intervention’. Please specify.:

We agree that the sentence is not clear. We changed to: “The development of high-quality and controlled trials with larger samples are needed to determine with greater certainty the interventions’ impact on parental feeding behaviors.”

4. Page 3; Line 45-46: “Adults are recognized as a major influence on children’s dietary patterns and can shape their preferences and food acceptance, guiding children’s eating behaviors during mealtimes, as positive role models, and through early exposure to different healthy foods [3, 4].” Very long and bit unclear sentence. Please adapt.:

We agree that the sentence is too long. We changed to: “Caregivers can shape children’s preferences and food acceptance through positive modeling, early exposure to different healthy foods and appropriate guidance of children’s eating behaviors [3, 4].”

5. Page 3; line 62: typo ‘parent’s actions to actions’. Please adjust.:

The sentence was corrected.

6. Page 4; line 68: correlations are reported here. Please reframe this part and clearly report about latest insight based on longitudinal and experimental data.:

The results reported in this sentence are related to a systematic review and meta-analysis about the influence of parental feeding practices on promotive and preventive child’s consumption behaviors. One of the inclusion criteria of the study is: “(…) studies had to utilize quantitative methods (surveys and experiments were both included), with statistical significance being reported”. We revised the phrase to be more clear: “On the other hand, structuring practices like modeling and food availability were positively correlated with both children’s healthy and unhealthy food intake: children tend to eat more of what their parents eat and what is available at home, independently of the type of food [12].”

7. 7. Page 5; line 86-87 “For children up to five years, interventions that target child feeding practices (e.g., repeated exposure, flavor-flavor learning, prompting, modeling) effectively increase fruit and vegetable intake and lasting up to 12 months [21].”: sentence unclear, please change for instance into ‘, with effects lasting…’:

The sentence was corrected: “For children up to five years, interventions that target child feeding practices (e.g., repeated exposure, flavor-flavor learning, prompting, modeling) effectively increase fruit and vegetable intake with effects lasting up to 12 months.”

8. The term ‘structure parental feeding practices’ is used throughout this piece, however, I am wondering whether this is a correct term, and whether ‘structure-related parental feeding practices’ would be a better term.:

This issue was raised by both reviewers. We changed to “structuring feeding practices” throughout the manuscript.

9. Page 14; line 151-152: “All the steps of the studies’ selection were conducted independently by two authors according to the defined inclusion/exclusion criteria.” What were these steps? This becomes clear from the flow diagram, but it may be helpful to shortly mention in the text that a two-phase procedure has been used in which first titles/abstracts and then full items were screened. And please mention for all these kind of sentences: “In the case of rating conflicts, both authors discussed and reached a consensus, helped by a third author” who these authors were by indicating their initials.:

We agree the selection process was not clearly described, so we changed the phrase according to your suggestions: “A two-phase procedure was used in which first titles/abstracts and then full articles were screened. Both steps of the studies’ selection were conducted independently by two authors (A.I.G. and L.B.) according to the defined inclusion/exclusion criteria. When disagreements occurred, a third author (A.I.P.) was consulted to reach a final decision by consensus.”. We added/revised the authors responsible for this specific tasks throughout the manuscript.

10. Page 21; line 278: “Five programs also focused other topics” = also focused on other topics?:

The sentence was corrected.

11. Were results similar found when studies having a high risk of bias were excluded? Mainly important for the last research question and meta-analysis.:

Thank you for raising this issue. The meta-analysis was performed considering all the studies selected (11 studies), including the studies with high risk of bias. In fact, because of the small number or studies selected and the variability of parental feeding practices measured in each study, it was impracticable to performed such analysis. This issue as discussed in the limitations of the study: “Also, the sample size made the exclusion of studies with a high risk of bias impracticable.”

12. General BCT use within the interventions has been reported. However, it might be more interesting to link BCT use to specific food parenting practices. This would also better align with the reported aim: line 405: “second, to identify behavioral change strategies that are chosen to change those parental practices”.: 

We agree that the analysis that you suggested is very interesting, that could add value to our manuscript, but unfortunately, the descriptions of the programs are not so detailed about the specific BCTs used to work on specific feeding practices.

13. Page 24; line 406: “and third, to analyze the impact of the interventions on those parental outcomes.” Reference to ‘those parental outcomes’ is a bit unclear. Please specify.:

We changed the sentence: “In this work, we addressed three main questions: first, to identify the parental feeding practices that are studied as outcome variables; second, to identify behavior change strategies that are chosen to change parental feeding practices; and third, to analyze the impact of the interventions on parental feeding practices outcomes”

14. Page 24; line 406-407: “Overall, we found that i) coercive controlling and structure parental feeding practices are commonly assessed”. Please also report in this main summary of findings which food parenting practices were not commonly assessed.:

Due to changes made in the discussion, we deleted the overall results related to the main research questions.

15. Page 25; line 414: “This small number of articles highlights the difficulty of finding programs assessing…”. I think this conclusion is somewhat weird. This probably highlights the fact that only few such intervention programs focused on the assessment of FPPs.:

Due to changes made in the discussion, we deleted this sentence.

16. Page 29; line 499-500: “Future programs should invest in raising parents' awareness of the use of this child-centered practices, and continue to work on the validation of instruments that appraise those specific parental behaviors. Recent intervention studies have often targeted responsive feeding. Please focus on implications of your main findings, that is, on feeding practices as outcome measures in interventions if I am correct.:

We agree with your suggestion and changed the sentence to better focus on the implications of our study: “Researchers should continue to invest in including child’s autonomy promoting feeding practices as outcome measures in interventions.”

17. Page 29: line 505-506: “Two programs with more BCTs (HomeStyles and EMPOWER) were described in several published articles about the program, which may have increased BCTs' global mean.” How is this possible? Given that the following procedure was reported on page 12, line 215-216 “Because different publications reported on the results from the same program, data were analyzed considering programs.: 

These two programs have more articles on the development of the program, with very detailed descriptions of the components of the program, increasing the possibility of finding more information about the BCTs used. However, because we had to reduce substantially the discussion, we decided to delete this sentence.

18. Page 25; line 412: “except for Food availability and accessibility when considering all follow-up measurements”. What does this mean? Please explain.:

We agree that these specific results about food availability/accessibility are not extensively discussed. So we add a paragraph with possible explanations about the results found.

19. Page 32; line 582: “In this review, we found an effect near significance on the availability of healthy food. These results suggest that changing children’s food environment regarding unhealthy foods is more challenging to achieve than increasing the availability of healthy foods.” Please further explain, because no effects were found on changing children’s food environment regarding healthy foods?:

Thank you for raising this issue. We agree with you: this is not a significant result so we deleted the discussion on those results.

---

## [Decision Letter · Decision Letter 1]

26 Mar 2021

PONE-D-21-02383R1

Changing parental feeding practices through web-based interventions: A systematic review and meta-analysis

PLOS ONE

Dear Dr. Gomes,

Thank you for submitting your manuscript to PLOS ONE. After careful consideration, we feel that it has merit but some minor issues raised by reviewer 2 need to be addressed before we can accept the paper for publication. Therefore, we invite you to submit a revised version of the manuscript that addresses the points raised during the review process.

We look forward to receiving your revised manuscript.

Kind regards,

Jane Anne Scott, PhD, MPH Grad Dip Dietetics, BSc

Academic Editor

PLOS ONE

Journal Requirements:

Reviewers' comments:

Reviewer's Responses to Questions

**Comments to the Author**

1. If the authors have adequately addressed your comments raised in a previous round of review and you feel that this manuscript is now acceptable for publication, you may indicate that here to bypass the “Comments to the Author” section, enter your conflict of interest statement in the “Confidential to Editor” section, and submit your "Accept" recommendation.

Reviewer #1: All comments have been addressed

Reviewer #2: All comments have been addressed

2. Is the manuscript technically sound, and do the data support the conclusions?

Reviewer #1: Yes

Reviewer #2: Yes

3. Has the statistical analysis been performed appropriately and rigorously? 

Reviewer #1: Yes

Reviewer #2: Yes

4. Have the authors made all data underlying the findings in their manuscript fully available?

Reviewer #1: Yes

Reviewer #2: Yes

5. Is the manuscript presented in an intelligible fashion and written in standard English?

Reviewer #1: Yes

Reviewer #2: Yes

6. Review Comments to the Author

Reviewer #1: Thank you for addressing the feedback provided, and I note your efforts to minimise the length and clarity of the article. A few further grammatical issues were noted:

Page 24, Line 405 – There is some confusion regarding objectives versus research questions – either frame these as questions (as per the introduction), or change your terminology here to ‘objectives’.

Page 25, Line 423 – Some feeding practices have been italicised while others have not – suggest italicising all feeding practices consistently for ease of reading

Page 27, Line 470 – ‘However, the Goal Setting was the BCT…’ – remove the first ‘the’

Page 28, Line 480 – Do you mean to say here that there should be a balance between collecting enough information to allow for tailoring and not burdening parents with excessive data collection / questions? The last part of the sentence is a little unclear and perhaps could be reworded (i.e. detailed records sounds like you are presenting them with records, whereas I think you mean collecting records from them?)

Page 29, Line 503 – Rather than ‘can be relevant’, perhaps ‘is suggested’

Reviewer #2: I would like to compliment the authors for the revisions they made. I have no further remarks and I think this piece will importantly contribute to the literature.

7. PLOS authors have the option to publish the peer review history of their article (what does this mean?). If published, this will include your full peer review and any attached files.

Reviewer #1: **Yes: **Chelsea E. Mauch

Reviewer #2: **Yes: **Junilla Larsen

---

## [Author Response · Author response to Decision Letter 1]

30 Mar 2021

Dear editor:

Thank you very much for the detailed and careful review. We made all the changes requested by the reviewers. The changes in the text are marked in blue, and below we answer all comments, indicating the changes made and providing additional explanations about the highlighted issues.

Journal Requirements:

Authors’ response: The reference list was reviewed. Although no major changes were performed (e.g., add or exclude references) since the last revision, in some cases, the article number/pages were missing. We have highlighted in blue all the changes made.

Reviewer #1: Thank you for addressing the feedback provided, and I note your efforts to minimize the length and clarity of the article. A few further grammatical issues were noted:

Page 24, Line 405 – There is some confusion regarding objectives versus research questions – either frame these as questions (as per the introduction), or change your terminology here to ‘objectives’.

We agree with your suggestion and changed these sentences to formulate them into questions

Page 25, Line 423 – Some feeding practices have been italicised while others have not – suggest italicising all feeding practices consistently for ease of reading

Thank you for noticing this issue. We revised and italicized all feeding practices in the text. We also noted that Food availability/accessibility practice was not always written in the same way, so we corrected the expression in the text to be according to the O’Connor categorization.

Page 27, Line 470 – ‘However, the Goal Setting was the BCT…’ – remove the first ‘the’

The sentence was corrected.

Page 28, Line 480 – Do you mean to say here that there should be a balance between collecting enough information to allow for tailoring and not burdening parents with excessive data collection / questions? The last part of the sentence is a little unclear and perhaps could be reworded (i.e. detailed records sounds like you are presenting them with records, whereas I think you mean collecting records from them?)

Thank you for your suggestion. We agree that the sentence can be improved, so we changed to: Researchers must decide on the right balance between collecting enough information to allow for tailoring and not burdening parents with excessive data collection and records.

Page 29, Line 503 – Rather than ‘can be relevant’, perhaps ‘is suggested’

We agree with your suggestion and changed the sentence.

Reviewer #2: I would like to compliment the authors for the revisions they made. I have no further remarks and I think this piece will importantly contribute to the literature.

Thank you very much for your contribute.

---

## [Editor Report · Decision Letter 2]

5 Apr 2021

Changing parental feeding practices through web-based interventions: A systematic review and meta-analysis

PONE-D-21-02383R2

Dear Dr. Gomes,

We’re pleased to inform you that your manuscript has been judged scientifically suitable for publication and will be formally accepted for publication once it meets all outstanding technical requirements.

Kind regards,

Jane Anne Scott, PhD, MPH Grad Dip Dietetics, BSc

Academic Editor

PLOS ONE
---

## [Editor Report · Acceptance letter]

19 Apr 2021

PONE-D-21-02383R2 

Changing parental feeding practices through web-based interventions: A systematic review and meta-analysis 

Dear Dr. Gomes:

I'm pleased to inform you that your manuscript has been deemed suitable for publication in PLOS ONE. Congratulations! Your manuscript is now with our production department. 

Kind regards, 

on behalf of

Dr. Jane Anne Scott 

Academic Editor

PLOS ONE